# Success Is Not the Entire Story for a Scientific Theory: The Case of the Phonological Deficit Theory of Dyslexia

**DOI:** 10.3390/brainsci12040425

**Published:** 2022-03-23

**Authors:** Pierluigi Zoccolotti

**Affiliations:** 1Department of Psychology, Sapienza University of Rome, 00185 Rome, Italy; pierluigi.zoccolotti@uniroma1.it; Tel.: +39-06-4991-7597; 2Neuropsychology Unit, IRCCS Fondazione Santa Lucia, 00179 Rome, Italy

**Keywords:** phonological theory, dyslexia, phoneme perception, picture naming, phonological awareness

## Abstract

In a recent paper, Share discussed four different “*Common Misconceptions about the Phonological Deficit Theory of Dyslexia*” and described this theory as “a *model of true scientific progress*” and a clear “*success story*”. In this note, I argue that at least part of the success of this theory is due to the lack of explicit predictions which make it very difficult (if possible) to test its predictions, and, possibly, falsify the theory. Some areas of pertinent research, including categorical phoneme perception, picture naming, and phonological awareness are summarized. Furthermore, two lines of research in which groups of researchers have attempted to formulate more explicit predictions are briefly outlined. It is concluded that, although much research has variously referred to the phonological deficit theory of dyslexia, the resulting large body of evidence presents a complex pattern of results which, in the absence of an explicit formulation of the theory, is extremely difficult to frame within a unitary interpretation. Overall, what seems needed is a theoretical formulation that, on the one hand, can account for the complex pattern of available evidence and, on the other hand, provide testable predictions for future research.

## 1. Introduction

In a recent paper in the special issue on “*Neurobiological Basis of Developmental Dyslexia*”, Share [1] discussed four different “*Common Misconceptions about the Phonological Deficit Theory of Dyslexia*”. In general, Share described the phonological theory as “*a model of true scientific progress*” and a clear “*success story*”. It is true that this proposal has gained a large consensus among the scientific and clinical community. Indeed, even some definitions of dyslexia variously refer to the role of deficits in phonological processing, potentially biasing the selection of participants (for a discussion, see [2]). However, it is possible to define dyslexia focusing on the reading deficiency only [3], e.g., as “*a persistent and unexpected difficulty in developing age- and experience-appropriate word reading skills*” [4].

In this brief opinion paper, I argue that success is not the entire story for a scientific theory; rather, the critical factor for a model/hypothesis/theory is to provide explicit predictions that can be tested in experimental research by the scientific community. In other words, what seems important in the evaluation of a scientific proposal is the possibility to falsify its statements through empirical research. The argument I will briefly develop here is that at least part of the success of the “*Phonological Deficit Theory of Dyslexia*” is due to the lack of explicit predictions which make it very difficult (if possible) to test, and, possibly, falsify the theory.

A first point that, in my opinion, needs to be considered is that there seems to be no clear and explicit description of the proposal that dyslexia is caused by a phonological deficit. In his title, Share referred to this as a “*theory*”; later in the paper, he referred to this idea as a “*hypothesis*”. Indeed, if one looks at the very large literature on the topic, this ambiguity is not occasional. Researchers referred to the idea that dyslexia is due to a phonological deficiency as a “*core deficit hypothesis*”, a model, or a theory. The term “*theory*” is more demanding as it should refer to a set of explicit and inter-related statements on the relationship between a set of phenomena. Is this the case for the phonological deficit theory of dyslexia? Despite the very large literature on the topic, there is no formulation of the theory that is sufficiently explicit to be directly testable. For example, Ramus and Ahissar [5] said that: “*The general consensus is that dyslexic individuals’ phonological representations are degraded (a hypothesis that has been formulated in a number of different versions, e.g., noisy, sparsely coded, underspecified, with poor spectral or temporal resolution ...)*”. What does it mean that phonological representations are degraded? In their quote, Ramus and Ahissar [5] pointed out that this may mean different things. Which one is the one proposed by the theory? The critical point here is that, if a hypothesis or theory is under-specified, the statement that individuals with dyslexia have impaired phonological representations risks representing a mere re-statement of experimental data. In their meta-analysis, Melby-Lervåg et al. [6] acknowledged the complexity of defining phonological representations and their role in acquiring reading: “*This is a difficult question to answer, because such representations can never be directly observed. Rather, their properties have to be inferred from patterns of behavior on phonological tasks. A reasonable assumption, however, is that a person lacking phonemically structured phonological representations would find phoneme awareness tasks difficult or impossible to perform*”. Thus, in this perspective, phoneme awareness (PA) tasks are considered critical measures of the phonological deficit. It is well known that as a group, children with dyslexia are impaired in phonological awareness tasks, such as deleting or adding a phoneme to a word or a pseudoword [6]. The critical point is to establish whether deficits in these tasks can be seen as a cause or actually a consequence of a reading deficit. This issue will be further discussed below (see Phonological awareness (PA)).

Share [1] himself did not provide an explicit description of the theory but instead relied more on metaphoric expressions. For example, he wrote: “*I argue that PA is only the “tip of the phonological iceberg” and that “deeper” spoken-language phonological impairments among dyslexics appear well before the onset of reading and even at birth*.” Elsewhere: “*Together with phonological awareness, this phonological ensemble was collectively labeled “phonological processing”.* In general, Share [1] seemed to make the point that there has been a considerable amount of research that provided consistent findings to provide strong support for the theory. For example, he said, “*The torrent of converging research findings came to the boil in the final decade of the twentieth century with numerous independent reviews and meta-analyses of the research literature …, spilling over into the early years of the 21st century*”. Furthermore, Share noticed that the popularization of the theory “*well beyond the bounds of the research community*” produced “*inevitable misunderstandings and misconceptions*”, the focus of his presentation.

Using metaphoric expressions such as “*ensemble*”, “*iceberg*”, or “*torrent*” represents an effective communicative means. However, it seems clear that this does not substitute for an explicit formulation of the theory. Below, I will provide a few examples of areas in which the lack of explicit predictions impedes a thorough evaluation of the theory. Accordingly, the goal of the present brief note is not that of a comprehensive evaluation of the theory (which would require a more detailed analysis of the whole body of relevant evidence). Rather, I will focus on some critical research areas to highlight the interpretive difficulties that arise from the fact that the theory is under-specified.

The phonological theory of dyslexia seems to rest on the idea that a number of phonological defects actually precede the difficulty encountered by the children when learning to read. In Share’s words: “*a steady accumulation of convergent findings pointed to subtle sub-clinical abnormalities in preliterate spoken language among dyslexics*.” On the one hand, the idea that these sub-clinical abnormalities precede the literacy disturbance appears important in that it places the reading disturbance within a broader linguistic perspective; on the other hand, the fact that the supposed deficits are “*subtle*” and “*sub-clinical*” makes it difficult to exactly predict which deficits are expected under which conditions, and which failures to detect deficits would run against the theory. In other words, there is a risk of circularity such that subtle deficits may be present, thus confirming the theory, but they may also be difficult to detect (because of their subtlety) so that their absence does not actually disconfirm the theory. Share quoted a number of signs, including “*word-finding and word-retrieval problems, difficulties recalling addresses, telephone numbers, the days of the week or months of the year, foreign names and places, early speech production and later articulation difficulties especially with phonologically challenging material such as tongue-twisters*.” Clearly, these are descriptive examples impinging on very different linguistic processes and do not represent explicit predictions. It would be hard to imagine that the theory is disconfirmed by the failure to find any of these putative deficits.

As stated above, the literature on linguistic correlates (or antecedents) of dyslexia is very large and a systematic summary of this body of research goes beyond the scope of the present note. However, it seems clear that along with “*positive*” findings, there are also “*negative*” findings. In a systematic review, Ramus and Ahissar [5] emphasized the importance of considering normal performance along with pathological performance in the interpretation of the phonological deficit theory of dyslexia. These authors noted for example that “*the most obvious prediction of the degraded phonological representations hypothesis should be that dyslexic individuals speak differently. However, there is very little evidence for that*.” Below, I briefly critically examine some research themes considered critical in the evaluation of the phonological deficit theory of dyslexia.

### 1.1. Categorical Phoneme Perception

In their review, Ramus and Ahissar [5] stated that “*the degraded phonological representations hypothesis should predict wide-spread disruption of the categorical perception of phonemes*”. By contrast, the evidence reviewed by these authors indicates that results are very variable, with studies reporting deficits in phoneme perception only in a sub-group of children or in some specific conditions and others finding no actual group difference. A similar pattern emerges when the results refer to the detection of phonemes under noise (a condition expected to enhance the sensitivity of the task); again, the results are mixed with different outcomes. Ramus and Ahissar [5] concluded that “differences are found only at very low signal/noise ratios … or under additional constraints (e.g., under high but not low presentation rate …).” Similar, inconsistent results have been reported in research carried out after this review. Thus, Berent et al. [7,8] reported moderate deficits in some (although not all) phonetic categories. In my opinion, the critical point here is that there is no clear way to determine whether these findings are consistent or not with the phonological deficit theory of dyslexia. Notably, the fact that not all conditions yield deficits in children with dyslexia is not in itself a limitation. One may envisage that some deficits will be present under specific conditions. However, the point is exactly that: to the best of my knowledge, there is not an explicit prediction of which should be the conditions under which a deficit of phoneme discrimination is expected and those in which it is not. In this way, one is left with the idea that, whenever deficits are present, they are supportive of the theory; whenever they are absent, they may have been too subtle to catch and thus they do not disconfirm the theory. To repeat this point for clarity, the results summarized by Ramus and Ahissar [5] are not necessarily incompatible with the theory; rather, it appears that the theory is not specified enough to be systematically tested. Thus, both positive and negative evidence may accumulate but there is no straightforward way to know whether it supports or runs against the theory.

### 1.2. Picture Naming

Another research area generating a host of mixed results is that concerning picture naming. Share [1] listed confrontation (picture) naming within the set of tasks mapping phonological processing. However, the pattern of evidence is complex. For example, it has been reported that children with dyslexia are less accurate in their naming of pictures [9,10], particularly if they are of low frequency or polysyllabic [10]. This finding has been taken as an indication of the difficulty in retrieving the phonological codes of known pictures, in keeping with the phonological deficit theory of dyslexia. However, subsequent work has proposed that the deficit may be subtle [11,12] and tends to vanish with age [13,14]. Furthermore, quite different results have been reported when vocal reaction times (RT) in naming pictures have been examined. In principle, this paradigm represents a particularly sensitive measure of individual performance; however, studies have generally failed to report a group difference in vocal RTs between individuals with and without a reading disorder [15,16,17,18]. Again, it seems difficult to frame these results within the phonological deficit theory of dyslexia. In the case of null results, ad hoc interpretations can be advanced. For example, Araújo et al. [15] proposed that the deficit in naming pictures might have been compensated by adult dyslexics (for a similar position, see [11]). Trauzettel-Klosinski et al. [17] proposed that reading rests on a phonological/orthographic pathway while a separate, and spared, visual pathway is used for picture naming. In a similar vein, in a study with my group, we proposed that dyslexia is specific for orthographic materials and does not necessarily extend to non-orthographic tasks, such as picture naming [16]. The absence of explicit predictions in the theory makes it impossible to evaluate these alternative proposals in a unitary fashion. For example, the idea that deficits in picture naming vanish with age is plausible but due to the lack of explicit predictions, it remains as a *post hoc* interpretation. Overall, it seems over-reaching to include, without further qualifications, picture naming among the tasks which reliably confirm the phonological deficit theory of dyslexia as done by Share [1].

### 1.3. Rapid Automatized Naming (RAN)

A closely related but distinct line of research is that which goes under the name of rapid automatized naming (RAN; [19] Denckla and Rudel). In this case, the child is requested to name in sequence a small set of targets (such as patches of color, objects, or numbers). Here, the evidence that children with dyslexia are slower in carrying out this task in comparison to control children is quite consistent (for a review, see Kirby et al. [20]). However, the interpretation of these findings is open to debate. Particularly critical for a phonological interpretation of the deficit in RAN is the experimental observation that group differences are specific for conditions in which the child is asked to name stimuli in sequence. By contrast, most studies found that when the same stimuli are presented singularly, the group differences are much smaller or absent (e.g., [21,22,23,24]). Furthermore, unlike performance in the classical RAN situation, performance in naming single targets is not (or minimally) related to reading performance (e.g., [24]). Note that if children would fail in RAN tasks because of impaired access to phonological representations, one would not expect a striking difference between single and multiple displays.

It may be added that several studies have reported dissociations with children showing deficits in RAN but not phonological awareness (PA) tasks and vice versa, an observation which is at the base of the double-deficit hypothesis of dyslexia [25]. This pattern has been extensively studied across several different orthographies (e.g., [26]). Results generally indicated that the partial dissociation between RAN and PA tasks holds for both opaque and transparent orthographies (e.g., [27]), the impact of these two predictors being generally stronger in the former as compared to the latter orthographies [26].

Overall, data on multiple-display RAN tasks indicate a consistent impairment in children with dyslexia. However, due to the absence of impairment with single display presentations, the nature of the disturbance does not appear consistent with a phonological interpretation but requires the contribution of other factors (for a discussion on this point, see also [28]). More generally, nowhere in the description of the phonological deficit theory of dyslexia is it anticipated that picture naming deficits should be present for multiple but not for single displays. Furthermore, individual differences in RAN tasks are reported as independent from individual differences in PA tasks [25], a finding difficult to reconcile with the idea that both deficits can be explained by a single phonological deficit.

### 1.4. Phonological Awareness

Over the years, phonological awareness (PA) has been an area of great debate. Furthermore, misconceptions 1 (“The phonological deficit hypothesis of dyslexia (PDH) is just about deficits in phonemic awareness (PA) and, consequently, is merely a circular “pseudo-explanation” or epiphenomenon of reading difficulties”) and 2 (“Deficits in PA (and other reading-specific skills such as pseudoword naming) are circular or pseudo-explanations of reading difficulties”) described by Share [1] also relate to this specific question. In PA tasks, the participant is asked to carry out operations such as deleting or adding a phoneme to a word or a pseudoword. It is well known that as a group, children with dyslexia are impaired in this type of task (for a meta-analysis, see Melby-Lervåg et al., [6]). However, it should also be added that if one takes into consideration individual cases, several children do not show a deficit in this type of tasks [25], an effect which may be particularly marked in transparent orthographies (e.g., [29]). Over the course of years, a debated point has been to establish whether deficits in these tasks can be seen as a cause or a consequence of a reading deficit. This question goes back to the observation in the late 1970s by Morais et al. [30], that illiterate individuals were not able to add or delete a phoneme from a non-word while individuals with even minimal, rudimentary knowledge of reading were rather successful. Accordingly, from this and related research (e.g., [31]), it may be concluded that PA is obtained through formal exposure to print, a conclusion which raises the question of circularity (considered in the Misconception 2 of Share).

Share [1] acknowledged that “*phonemic awareness is not a universal and emergent linguistic capability but is best categorized as a reading subskill*.” How does one avoid then the risk of possible circularity connected with the interpretation of performance in PA tasks? The critical point in Share’s analysis is the proposal that PA must be seen as a proximal precursor (or risk factor) of reading not as a (distal) cause. The distinction between proximal and distal causes has been important in the genesis of cognitive neuropsychology. In general, cognitive modeling has been concerned with developing architectures that describe how some proximal factors and their interactions may account for a given target behavior [32]. According to Coltheart [32], factors should be seen as “*distal*” if they affect behavior only indirectly by influencing the proximal factors in the model. Notably, cognitive skills are neither proximal nor distal as such; rather, this distinction refers to the way in which one looks at a given cognitive factor (for a thorough discussion, see [32]).

Personally, I think that framing PA performances in proximal terms is an interesting proposal with potential intriguing implications and developments. Still, it should be observed that this can hardly be seen as a clarification of a common misconception about the role of PA. Indeed, the up-to-date most extensive review of the literature on PA (also quoted by Share) concluded that “*the present meta-analyses are consistent with the claim that the development of phonemically structured phonological representations is one critical foundation for learning to read successfully in an alphabetic script … and that a failure to develop such phonemically structured phonological representations is a principal cause of the difficulties in learning to read experienced by children with dyslexia …*” [6]. Thus, these authors directly proposed that performance in PA tasks can be seen in causal terms in explaining dyslexia and no mention of the distinction in terms of the proximal-distal dimension is presented in the paper. Put in other terms, seeing PA tasks in proximal terms, as proposed by Share [1], is not a clarification of the past but offers a new (and potentially interesting) way of looking at these tasks.

I think it is important to spell out the implications of PA when seen in a proximal perspective. Proximal factors come to life within cognitive architectures of a given behavior. Particularly, one expects that a cognitive architecture makes explicit the nature of the relationship between the predictors and the target behavior. A few authors have proposed descriptive sketches of the processes involved in reading (e.g., see Figure 1 in Vellutino et al. [33]) which consider PA tasks. However, there does not seem to be general agreement over a single and explicit cognitive architecture which includes PA tasks as proximal predictors. As stated above, Share referred to a variety of tasks/conditions stating that “*together with phonological awareness, this phonological ensemble was collectively labeled “phonological processing*“. Yet, the “*ensemble*” term points to the collective and unstructured nature of this set of predictors. Thus, seeing PA in proximal terms has the direct implication that one expects the nature of the relationship with reading to be spelled out more clearly as part of a structured cognitive architecture. It appears that this enterprise still needs to be fulfilled.

A second implication of seeing PA in proximal terms is that PA measures can be examined along with other proximal predictors although they should not be directly compared with measures aimed to capture long-range (distal) relationships with reading. There is a very rich tradition in the research on dyslexia as to examine possible mechanisms which represent the basis for the reading disturbance (and the articles on the Special Issue on “Neurobiological Basis of Developmental Dyslexia”, are largely devoted to this aim). For example, studies have examined the possible contribution of impairments in magnocellular processing [34], in temporal processing [35], or in cerebellar functioning [36]. Within the framework proposed by Coltheart [32], these studies deal with “*distal*” predictors. On the one hand, the actual relationship why a failure in a task such as balancing a rod (or detecting the movement of a grating) should impair reading is not explicitly spelled out (i.e., they are not seen as proximal predictors); however, on the other hand, they do not suffer from the close circular relationship with reading which affects PA tasks. Thus, it is hard to see how learning to read influences the acquisition of skills such as balancing a rod or detecting the presence of movement.

An implication of this reasoning is that in predicting dyslexia, proximal and distal predictors serve different purposes and should not be mixed in a direct comparison. However, in various studies, PA tasks have been used along with measures aimed to capture underlying basic mechanisms. For example, Ramus et al. [37] compared a series of different theories of dyslexia, including the phonological theory, the magnocellular (auditory and visual) theory, and the cerebellar theory. A very extensive battery of tests mapping various basic mechanisms was used. Critically, for measures of magnocellular or cerebellar performance, there was no manifest, intrinsic relationship with reading tasks (i.e., these tests should be considered as distal predictors, according to [32]). Thus, it is difficult to see how learning to read directly affects performance in bead threading or repetitive finger-tapping (taken as indicators of cerebellar functioning); similarly, it does not seem likely that learning to read influences coherent motion detection (taken as an indicator of magnocellular functioning). Indeed, this latter statement has also been the object of direct research; thus, it has been reported that a rehabilitation training of reading has no effect on the threshold in the detection of coherent motion [38]. In other terms, examining these predictors represents a search for basic mechanisms putatively underlying problems in the acquisition of reading. Although the close relationship with reading is left underspecified (i.e., they are not framed as proximal predictors), they do not suffer from an intrinsic relationship with reading. By contrast, the phonological theory was tested largely based on PA tasks for which, as specified above, there is a well-known and explicit link with reading (which makes it reasonable to see them as proximal predictors, as proposed by Share [1]). Unsurprisingly, Ramus et al. [37] reported that adults with dyslexia were most affected in a phonological index which included PA tasks and much less in magnocellular or vestibular tasks; however, this conclusion directly compares the effects of distal and proximal factors (even if the authors did not directly use these terms).

In a different study, Saksida et al. [39] aimed to compare three different causes of dyslexia: a phonological deficit, a visual stress deficit, and a reduced visual span (not examined here). Again, this study envisaged the direct comparison between PA tasks (such as a phoneme deletion task and a Spoonerism test) with measures aimed to detect possible mechanisms underlying dyslexia. Thus, visual stress was measured by presenting a high spatial frequency grating and asking children questions concerning whether lines appeared distorted, in movement, etc. Saksida et al. [39] concluded confirming the role of a phonological deficit and not of a visual stress one. Again, it seems important to underscore the different nature of the tasks being compared. In the case of PA tasks, the question is not if they are related to reading. As stated above, PA performance is tuned through literacy acquisition; therefore, using these measures as if they were distal predictors of reading seems inappropriate. By contrast, there is no clear reason to think that learning to read has a direct (or even reciprocal) interaction with performance on the visual stress test used.

A third implication of seeing PA tasks as proximal predictors is that one may want to consider whether deficits in basic sensory mechanisms underlie the observed impaired performance in PA tasks, hence providing a comprehensive theoretical proposal of the phonological hypothesis. Some authors have taken this perspective. For example, Tallal [40] has proposed that phonological deficits are due to an underlying deficiency in perceiving fast acoustic changes (for a critical evaluation of the evidence on this hypothesis, see [37,41]). An alternative proposal has been advanced by Goswami [42]; the temporal sampling framework proposal envisages that the inefficient phonological processing present in dyslexia is due to underlying deficits in temporal sampling and inefficient phase locking. Particularly, there would be deficits in amplitude envelope rise times at relatively slow temporal rates, those affecting the detection of speech rhythm and prosody [42]. A comprehensive evaluation of these two hypotheses goes beyond the aims of this note. Here, I limit myself in noticing that the phonological deficit theory of dyslexia leaves unspecified whether derangement of a basic sensory mechanism is hypothesized at the origin of the reading deficit. The idea that PA performance should be viewed in terms of proximal predictors makes it clear that some reference to a more basic mechanism is in order. Although there are different proposals to this aim, there does not seem to be yet a consensus as to which is the basic mechanism underlying impaired PA performance. However, this is an important goal if a phonological deficit theory of dyslexia wants to provide a comprehensive account of the reading deficit.

To summarize, seeing PA tasks as proximal predictors may avoid circularity in the causal interpretation of the disturbance, as proposed by Share [1]. However, an important implication of this perspective is that one should be able to articulate a cognitive architecture that allows appreciating the role of PA tasks in the interpretation of reading deficits, i.e., to frame PA as a proximal factor. Another implication of this perspective is that PA should not be used along with predictors aimed to capture basic mechanisms underlying reading (i.e., distal factors) in a comparative evaluation of which deficit best accounts for the reading deficit. Furthermore, emphasizing the proximal nature of PA makes it clearer that a comprehensive formulation of the theory would require a hypothesis about the basic sensory mechanism underlying deficient performance in PA tasks, i.e., the distal factors influencing PA.

Finally, it should be added that these comments apply to the population of children with dyslexia as a group and this is certainly an over-simplification. It is well established that there are large individual differences in PA tasks, with several children behaving well within normal limits (e.g., [25]). A comprehensive phonological deficit theory of dyslexia should also account for these individual differences (as well as for the partial dissociation with RAN tasks quoted above [25]).

## 2. Lines of Research Which Developed Original Predictions to Test the Phonological Deficit Theory of Dyslexia

As described above, several authors have framed their results within the phonological deficit theory of dyslexia even in the absence of explicit predictions. However, in a limited number of instances, groups of researchers have attempted to formulate more explicit predictions derived from their understanding of the theory. Considering these approaches and the related findings is interesting and informative for an evaluation of the phonological deficit theory of dyslexia. Two such lines of research will be briefly outlined below.

In a series of papers, Berent et al. [7,8,43] distinguished between a phonological grammar, an amodal system of rules computing the structure of linguistic primitives, and a phonetic interface that allows mapping the abstract phonological patterns into specific speech sounds. Accordingly, one should look at the deficiencies shown by individuals with dyslexia clearly distinguishing the phonological and phonetic levels of processing. Particularly, Berent et al. [7,8,43] proposed that seeing the dyslexia deficit in terms of impaired phonological representations, one would expect individuals with dyslexia to be impaired in tasks mapping on the phonological grammar.

In studies on individuals with dyslexia speaking either Hebrew [7,43] or English [8], Berent et al. contrasted performance on various tasks mapping on the phonological or phonetic levels of processing. One critical characteristic of phonological rules is their productivity, i.e., the possibility to extend them to new items. For example, one universal constraint on the syllabic structure is that syllables with small sonority clines are seen as ill-formed and are expected to be recoded or “repaired” (e.g., “lbif” → “lebif”). The capacity of making such repairs represents an active indication of a spared phonological grammar. Using a variety of experimental conditions, Berent et al. [7,8,43] observed that both Hebrew and English individuals with dyslexia were able to generalize a grammatical phonological rule, a finding in sheer contrast to the predictions. They also tested the same groups of individuals on phonetic identification and discrimination tasks. Results were more variable, indicating small deficits in individuals with dyslexia in some categories but not others. For example, English individuals with dyslexia were impaired in identifying the /ɔ/-/ʊ/ vowel continuum but not those related to /ta/-/da/, /ba/-/pa/, or /æ/-/e/ continua [5]; none of these latter contrasts generated group differences in the case of the discrimination task. Somewhat more positive results were reported for Hebrew-speaking individuals [4,35]. Still, it seems clear that phonetic deficits were generally subtle and highly variable for different contrasts and from study to study. These findings add up to the evidence reviewed by Ramus and Ahissar [5] and briefly summarized above, indicating small and variable difficulties in categorical phoneme perception.

Overall, the studies carried out by Berent et al. [7,8,43] indicate that individuals with dyslexia show an entirely spared phonological grammar, a finding which the authors described as incompatible with the phonological deficit theory of dyslexia.

Another significant line of research aimed to characterize the nature of the putative phonological impairment in dyslexia has been developed in a series of studies by Ramus, Szenkovits, and colleagues [44,45,46]. These authors tackled three different main research questions. First, they examined the putative locus of the phonological deficit [44,45]. Second, they tested the phonological similarity effect in conjunction with dyslexia [45]. Third, they examined phonological grammar in dyslexia in relationship with specific characteristics of a language [46].

The first set of experiments was exploratory in nature [44,45]. They tested whether the nature of the impairment in phonological representations, implied in the phonological interpretation, involved the lexical or the sublexical level be comparing words and nonwords; they also contrasted the input or output levels using repetition and discrimination tasks, with the idea that the former involves both input and output levels while the latter only the input representations. Results indicated that the performance of dyslexics was impaired in all experimental conditions, indicating no specific locus for the putative deficit in phonological representations.

Second, Ramus and Szenkovits [45] examined the phonological similarity effect, i.e., the tendency for a more difficult recall of sequences of phonologically similar words. Ramus and Szenkovits [45] hypothesized that, if the phonological representations of individuals with dyslexia are impaired, they should show a particularly defective performance in remembering phonologically similar sequences. Results confirmed the presence of a phonological similarity effect: sequences of nonwords differing by just one phonetic feature (e.g., [taz]–[taZ]) were more difficult to remember than nonwords differing for three phonemes (e.g., [taz]–[gum]). However, contrary to the prediction, the size of the phonological similarity effect was quite similar between individuals with dyslexia and control readers. The authors concluded that “*these results fail to confirm the predictions of the “degraded phonological representations” hypothesis*” [45].

Finally, Szenkovits et al. [46] also tested for the possible defect in phonological grammar in individuals with dyslexia, a line of research parallel to that of Berent et al. [7,8,43] described above. Particularly, they were interested in the acquisition of phonological assimilation rules in relationship with a specific language, French, which shows voicing, but not place assimilation. They tested the capacity of individuals with dyslexia and control readers to produce systematic variations in their production of words as well as their perceptual capacity to compensate for such variations to comprehend the target words. Results generally confirmed the language-specific nature of the assimilation rules. In the speech production task, speakers produced voicing assimilation but limited to an assimilatory context; in a task of word detection in sentences, participants compensated for assimilation more when stimuli occurred in native assimilatory contexts and more for voicing than for place. Critically, both in the production and perception tasks, the pattern of findings was identical for individuals with dyslexia and control readers. The authors concluded that “*this runs against any variant of the degraded phonological representations hypothesis that involves a degradation of voicing or place features*.” Furthermore, they stated that “*the results we obtained here suggest that dyslexic individuals have both intact phonological representations and a normal phonological grammar, or that if their phonological representations or grammar are affected, they are so subtly affected as to seriously diminish the plausibility of such degradation significantly affecting reading acquisition*” [46].

Overall, these two groups of researchers have tried to derive explicit and testable hypotheses to support or disconfirm the phonological deficit theory of dyslexia. Both investigated the question of phonological grammar although from a partly different perspective. Berent et al. [7,8,43] focused on the amodal, abstract quality of phonological grammar while Szenkovits et al. [46] examined the capacity of individuals to apply assimilation rules in relationship with a specific language. Still, the results were very consistent indicating in all cases spared phonological grammar in individuals with dyslexia speaking Hebrew, English, and French. Additionally, one may add that similar results were also reported in a similar study on Dutch children [47], in keeping with their reliability. In the case of the phonological similarity effect, no group difference was found comparing individuals with and without a reading disorder. How can these findings be framed with the phonological deficit theory of dyslexia? Clearly, one might see these results as inconsistent with the theory and, to a different degree, this is what the authors of the studies have done. However, it is not entirely clear whether these negative results should be taken to invalidate the theory. As stated above, the formulation of the phonological deficit theory of dyslexia has not been sufficiently explicit as to say whether these tests were the critical ones to verify the theory. For example, no clear distinction between a phonological grammar and a phonetic interface is present in most literature on the phonological theory of dyslexia. In other terms, the quoted evidence seems inconsistent with the theory but is it an actual test of its predictions?

Note that even negative results may not necessarily lead to a rejection of the theory. For example, Ramus and Szenkovits [45] made the hypothesis that the phonological representations of individuals with dyslexia are spared but they may be difficult to access. Accordingly, the deficit may be particularly sensitive to the characteristics of the task (e.g., in terms of short-term memory or attentional requirements) under which phonological representations are tested. Alternatively, as we have seen, it has been proposed that dyslexia is not associated with a deficit in phonological representations but to one in the phonetic interface connecting abstract phonological patterns into speech sounds [7,8,43]. Establishing whether one of these alternatives offers a plausible and comprehensive interpretation of the currently available evidence on the issue goes well beyond the aims of the present brief note. Here, I just wanted to illustrate how, in the absence of an explicit formulation of the phonological deficit theory of dyslexia, it is extremely difficult (or perhaps impossible) to establish its soundness.

## 3. Conclusive Remarks

As stated by Share [1], in the last decades, much research has variously referred to the phonological deficit theory of dyslexia. In this brief note, I argue that this has occurred in the absence of an explicit formulation of the predictions of the theory. Rather, the theory has been assumed (not tested) with the idea that its definite formulation would arise progressively as an emerging property of the accumulation of confirmatory findings. The problem with this approach is that the up-to-date large body of evidence presents a complex pattern of positive and negative results (both in terms of group differences and patterns of individual performances) and, in the absence of an explicit formulation, it is extremely difficult to decide which findings should be considered crucial in a critical evaluation of the theory. Even the present sketchy presentation makes it clear that a single, generic statement about the role of quality of phonological representations in reading falls short of significance in the presence of the available evidence.

If one wants to make a scientific advancement, what really seems needed is an attempt to make a theoretical formulation that, on the one hand, can account for the complex pattern of available evidence and, on the other hand, provide testable predictions for future research. It is my hope that this note may contribute to foster this process.

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
