# Peer review of "Success Is Not the Entire Story for a Scientific Theory: The Case of the Phonological Deficit Theory of Dyslexia"

_brainsci, 2022, doi:10.3390/brainsci12040425_

Round 1

Reviewer 1 Report

Success is not the entire story for a scientific theory: the case of the Phonological Deficit Theory of Dyslexia

In this opinion paper, the author claims that it is not yet adequate to view the phonological deficit theory of dyslexia as a successful theory for two reasons: (1) in most studies, explicit predictions based on this theory have not been made and, consequently, they remain untested; (2) when explicit predictions have been made, tests may not have been sensitive enough to test those predictions.

I found this a very stimulating text, well-written and well-organized. However, I am not totally convinced with the major organizing concepts of the discussion, namely the idea of “explicit predictions” and “lack of sensitivity of tests”. I also believe that the manuscript could increase its contribution by emphasizing the possible solutions for the problems that have been identified. I outline these concerns below.

1-EXPLICIT PREDICTIONS: The idea that hypothesizing deficits in categorical phoneme perception, picture naming, RAN and phonological awareness in dyslexia do not constitute explicit predictions is not clearcut to me. Saying something like “dyslexics have lower performance than controls in (any of these tasks)” seems totally testable and explicit, in the sense that one can easily build an experiment based on this.

1.1. One of the reasons why the author claims these hypotheses are not yet explicit enough is that these deficits seem to depend on moderating variables (specific circumstances or phonemes for categorical perception; specific ages for picture naming; presentation mode in RAN). Therefore, predictions should have been more precise. But then I ask: why should they have been more precise? Was there any theoretical reason related to the phonological approach allowing more precision in predictions?

1.2. In the case of phonological awareness, I see a similar problem: the author suggests the prediction is not explicit, but I cannot see why (saying that – at least pre-literate - children have lower PA skills than controls is something testable). The fact that tests on PA skills should be coupled with tests on the whole chain (sensory origin of PA deficit), or that PA tests should not be confronted with tests on distal causes like magnocellular problems does not make the hypothetical PA deficit difficult to test.

1.3. Could it be that, in section 1, the author means that the concept of phonological deficit is being operationalized at the task level (like categorical phoneme perception, picture naming, RAN and PA), without previous conceptualization of the components of the phonological system that potentially define the tasks, and the absence of reference to these components makes the theory incomplete and the operationalization imperfect? This seems to me like the key difference between research pointed out at section 1 vs. section 2. However, if this is the case, I do not understand the objections referred to in my comments 1.1 and 1.2.

1.4. Indeed, in section 2, the author claims that considering a phonological grammar and a phonetic interface makes the phonological theory more testable. Is it the key to testability? I wonder if it would not be interesting to start the paper by stating which attributes a theory should have to be testable.

2- LACK OF SENSITIVITY WHEN TESTING EXPLICIT PREDICTIONS: Research pointed out in section 2 is viewed as based on testable predictions. However, the null results that come out from these studies are not seen by the author as evidence that the theory is false, but, rather, as indices of lack of sensitivity in tests. I have two questions:

2.1. Why should we not see these null results as evidence that the theory is false?

2.2. What does it mean to have testable predictions tested with insensitive measures? Are we still dealing with testable predictions?

3-SOLUTIONS: Finally, I would like to read more about directions and solutions. The author mentions one or another, for instance, testing PA together with tests on sensory deficits that may lead to PA. However, I missed a global approach to the problem: How should the theory be reworked to be useful? I understand that the author is not committed to falsify the theory. Therefore, we need to “be unable to falsify it” in a proper way. How should this be done – considering phonological components, sensitive measures o these components and what else? Where does the suggestion made for testing PA together with sensory deficits stand here?

Author Response

Reviewer 1

Success is not the entire story for a scientific theory: the case of the Phonological Deficit Theory of Dyslexia

In this opinion paper, the author claims that it is not yet adequate to view the phonological deficit theory of dyslexia as a successful theory for two reasons: (1) in most studies, explicit predictions based on this theory have not been made and, consequently, they remain untested; (2) when explicit predictions have been made, tests may not have been sensitive enough to test those predictions.

I found this a very stimulating text, well-written and well-organized. However, I am not totally convinced with the major organizing concepts of the discussion, namely the idea of “explicit predictions” and “lack of sensitivity of tests”. I also believe that the manuscript could increase its contribution by emphasizing the possible solutions for the problems that have been identified. I outline these concerns below.

Response: I want to thank the reviewer for his/her kind comments on the text. Before answering to the specific points raised by the reviewer, I would like to make the general point that in developing the argument object of my opinion paper I am trying to move (using a metaphor), in a “narrow road”: the aim of the paper is not that of making an extensive evaluation of the PDH theory (which would have requested a considerably larger analysis and a more extensive rendition). More modestly, I wished to comment on the paper by David Share (“Common Misconceptions about the Phonological Deficit Theory of Dyslexia”) by raising the point that the PDH theory is largely underspecified and that this impedes evaluating the way experimental results fit with this theoretical proposal. So, in the text I generally restrained from taking definite conclusions in commenting the experimental data (see further comments below) but limited myself in noting the lack of specificity of the theory (in comparison to the idea put forward by prof. Share that “the popularization of the theory well beyond the bounds of research community” produced “inevitable misunderstanding and misconception of the theory”).  In re-reading the manuscript I realized that this point could have been made more explicitly; so, I added two sentences in the par in page 2 (“Using metaphoric expressions such…); I hope the new version proves clearer.

My hope was that my paper might raise in the (several) authors working within the PDH theory the awareness that it is time to go beyond broad (but generic) statements about the relevance of phonology in reading and try to make explicit predictions which take into account the already large body of (both positive and negative) evidence available in this area. I modified the last sentence of the manuscript to express better this concept.

1-EXPLICIT PREDICTIONS: The idea that hypothesizing deficits in categorical phoneme perception, picture naming, RAN and phonological awareness in dyslexia do not constitute explicit predictions is not clearcut to me. Saying something like “dyslexics have lower performance than controls in (any of these tasks)” seems totally testable and explicit, in the sense that one can easily build an experiment based on this.

1.1. One of the reasons why the author claims these hypotheses are not yet explicit enough is that these deficits seem to depend on moderating variables (specific circumstances or phonemes for categorical perception; specific ages for picture naming; presentation mode in RAN). Therefore, predictions should have been more precise. But then I ask: why should they have been more precise? Was there any theoretical reason related to the phonological approach allowing more precision in predictions?

Response: I apologize for not having be enough clear in my text. One expects a theory to explain all the currently available evidence and possibly make further testable hypotheses for future research.  In the case of the experiments quoted in 1.1. (e.g., “Categorical phoneme perception”), the general expectation (as stated by Ramus and Ahissar) is that “the degraded phonological representations hypothesis should predict wide-spread disruption of the categorical perception of phonemes”.  The evidence reviewed by these authors indicates that, by and large, results do not conform to this expectation.  One straightforward way to look at this is to reject the theory.  However, as stated above, this was not the goal of my comment, since I did not make a systematic review of the literature which would represent the necessary basis for such a definite conclusion. 

The focus of my note is in trying to underscore the fact that in order to maintain a theory when results do not fit expectations you need to spell out more clearly the theory to fully account for the empirical evidence available or, alternatively, abandon altogether the theory itself.  As stated above, I feel that my presentation is not detailed enough for this second goal (this point is spelled out more clearly in the new version of the manuscript).  Further, it is true that, despite several negative findings, the theory receives a large consensus in the literature as stated by Share in his paper.  Accordingly, my (selective and limited) goal was to underscore that, if one wants to maintain this theoretical formulation, predictions/expectations should be tuned to account for the actual findings reported in the literature.  In this sense, more precise predictions are needed not only because of theoretical reasons but because accumulated evidence does not easily fit into a generic statement of the theory (such as that by Ramus and Ahissar quoted at the beginning of this response).

1.2. In the case of phonological awareness, I see a similar problem: the author suggests the prediction is not explicit, but I cannot see why (saying that – at least pre-literate - children have lower PA skills than controls is something testable). The fact that tests on PA skills should be coupled with tests on the whole chain (sensory origin of PA deficit), or that PA tests should not be confronted with tests on distal causes like magnocellular problems does not make the hypothetical PA deficit difficult to test.

Response: Actually, the question I have discussed in the case of PA skills concerned more the nature of this measures than the fitness to the predictions of the theory.  There has been a large debate on how PA measures should be interpreted.  The focus of this part of the text was largely in commenting the implications of the proposal by prof. Share that PA measures should be seen in terms of proximal (not distal) predictors of reading.  So, it is indeed not difficult to test PA deficits, but it seems important to understand what the relevant findings mean.

1.3. Could it be that, in section 1, the author means that the concept of phonological deficit is being operationalized at the task level (like categorical phoneme perception, picture naming, RAN and PA), without previous conceptualization of the components of the phonological system that potentially define the tasks, and the absence of reference to these components makes the theory incomplete and the operationalization imperfect? This seems to me like the key difference between research pointed out at section 1 vs. section 2. However, if this is the case, I do not understand the objections referred to in my comments 1.1 and 1.2.

Response:  Yes, I generally agree with the reviewer’s comments.  There has been a lot of research on several tasks loosely falling on the phonological category but as stated by the reviewer, it seems that the conceptualization of the phonological basis of these tasks has been incomplete. The overall goal of the paper is to raise the awareness of researchers working in this area of research about the importance of more clearly define the processes underlying this complex set of data and make explicit predictions tuned to the different tasks/conditions/processes.  At the same time, I hope I made it clear in my previous responses why I do not see this perspective as inconsistent with what written with regard to the various tasks (considered in 1.1. and 1.2).

1.4. Indeed, in section 2, the author claims that considering a phonological grammar and a phonetic interface makes the phonological theory more testable. Is it the key to testability? I wonder if it would not be interesting to start the paper by stating which attributes a theory should have to be testable.

Response: It is somewhat difficult to state whether “considering a phonological grammar and a phonetic interface makes the phonological theory more testable”.  Personally, I feel that this is because the phonological deficit theory is largely underspecified. So, on the one hand one could say that making these predictions is quite reasonable (and this should lead to a rejection of the theory as experimental evidence was largely negative).  On the other hand, one could say that the theory did not explicitly state that there should be differences in, say, phonological grammar.   So, one could argue that operationalizing the theory in terms of a phonological grammar and a phonetic interface was not actually envisaged by the original formulation of the theory.  Which position is correct?  My personal opinion is that, in spite of its popularity and success, the theory is underspecified.  So, it is not possible to decide which of the two alternatives should be considered.

It is certainly intriguing to see that, in spite of negative evidence, the researchers who carried out the relevant studies did not themselves directly reject the theory but took a more articulated position.  Ramus for example maintained that the theory should be held (citing reasons that are difficult to summarize here).  Berent seems to have taken the position that the theory should be restated in terms of the phonetic interface (but please note that also the results concerning this aspect were largely negative). Individual positions are certainly legitimate. However, they are not necessarily relevant in the systematic evaluation of a theory.  I made these examples mostly because they testimony the difficulty of researchers in abandoning a theory in spite of contrary evidence (in keeping with the idea described in the Share’s paper is that there is a large consensus on this general approach/theory).

2- LACK OF SENSITIVITY WHEN TESTING EXPLICIT PREDICTIONS: Research pointed out in section 2 is viewed as based on testable predictions. However, the null results that come out from these studies are not seen by the author as evidence that the theory is false, but, rather, as indices of lack of sensitivity in tests. I have two questions:

2.1. Why should we not see these null results as evidence that the theory is false?

2.2. What does it mean to have testable predictions tested with insensitive measures? Are we still dealing with testable predictions?

Response: I apologize for not being entirely clear on this point. As stated above, in a comprehensive systematic review of the literature on the PDH theory it will be important to reach a conclusion on how to interpret results such as the lack of deficits of children with dyslexia in tasks of phonological grammar.  Certainly, null results can be interpreted as evidence that the theory is false as stated in point 2.1. (and I have mentioned this possibility in lines 424-426 of the original version of the manuscript).  The more specific point to which I limited myself is that, in an absence of an explicit formulation of the PDH theory, it is difficult to see whether experiments such as those carried out by Berent or by Szenkovits and Ramus are actually critical for testing the theory.  I expressed this idea in the following way: “As stated above, the formulation of the Phonological deficit theory of dyslexia has not been sufficiently explicit as to say whether these tests were actually the critical ones to verify the theory.”

I should add that the experiments carried out by these authors were actually effective in generating the expected pattern (e.g., the phonological similarity effect or the presence of voicing assimilation limited to an assimilatory context in French).  Thus, there does not seem to be a problem of sensitivity in these measures. I have carefully reread the text to control for this and made some changes for the sake of clarity. In particular, I have noticed that I made an imprecise reference to the concept of sensitivity (in previous line 433) which I have deleted it in the new version.  As stated above, what I think is missing the theoretical link to the PDH theory not so much the sensitivity of the measures.  I apologize for this confusion.  I hope that the new version proves clearer.

3-SOLUTIONS: Finally, I would like to read more about directions and solutions. The author mentions one or another, for instance, testing PA together with tests on sensory deficits that may lead to PA. However, I missed a global approach to the problem: How should the theory be reworked to be useful? I understand that the author is not committed to falsify the theory. Therefore, we need to “be unable to falsify it” in a proper way. How should this be done – considering phonological components, sensitive measures of these components and what else? Where does the suggestion made for testing PA together with sensory deficits stand here? 

Response: The points raised by the reviewer are all quite important. However, I feel that the aims of my short note fall short of them.  I feel that this is a goal for people committed to the phonological theory (and Share’s paper makes it clear that there are quite a significant number of researchers in this position).  By contrast, I do not feel qualified to make such as theoretical statement.  The (much more limited and focused) aim of my note is to underscore that we should not focus on the success and large consensus of a theory but try to make it explicit and testable. Paradoxically, as stated by the reviewer, it seems that in the current state of affairs is not possible to formulate predictions that (if unfilled) would falsify the theory.  This may be reassuring for some, but it seems an important limitation of the overall approach.  Again, my hope is that my note would raise the awareness of this limitation of the current formulation of the theory.

Reviewer 2 Report

see attached file

Author Response

Reviewer 2

Comments and Suggestions for Authors

The paper is very clear and presents arguments that are timely and needed to move the field forward. I have only a few suggestions that I would like Professor Zoccolotti to consider:

  1. The term dyslexia is used assuming that it means the same thing for different readers. As (1) some definitions include mention of phonological deficits and others do not, and (2) some studies use participant selection criteria that limits participants to those with a phonological deficit, I think it would also be useful for the internal consistency of this paper to provide a definition of dyslexia that does not include phonological deficit. I understand definitions come with a whole new set of issues, but I believe anchoring the arguments against the authors definition would still be valuable.

Response: I thank the reviewer for this useful suggestion.  I have included a definition of dyslexia (which focusses on the reading deficiency only) as well as some references of papers which discuss this question.

  1. In few places I read the argument that it is well known that individuals/children with dyslexia are impaired in phonological awareness tasks (pp. 2 & 4 at least). This is true only if the sample was selected on the basis of phonological deficit. When samples are selected on the basis on poor reading accuracy or efficiency, the dyslexic groups may still be poorer than the control group but when individual level information is available, up to half of them do not show phonological deficit. These statements should be qualified by "as a group" or "many, but not all" or something of that kind - if not, it perpetuates the misperception of universality of poor performance in these tasks. I think this is also true with categorical perception tasks - when distributions of scores are reported, there is a large overlap between groups with many individuals with dyslexia performing at normal levels. 

Response: I would like to thank the reviewer for this observation. I entirely agree that deficits in phonological awareness (as well as categorical perception tasks) show large individual differences and that these are particularly relevant.  In the first version of the manuscript, for the sake of the brevity I left this important point aside.  I have modified this point in the revision (inserting a sentence at lines 250-252, page 5, of the revised paper), and I hope the new text proves clearer.

  1. In terms of RAN deficits, it may also be worth mentioning that several studies have found a group that has RAN but no PA deficits --> if both are explained by phonological deficit, we need two independent deficits or a well-developed severity argument (and I am not aware of one).

Response: This is an important point which I did not make in the first version of my manuscript.  I have tried to incorporate these observations in the present revision (inserting a sentence at lines 227—240, page 5, of the revised paper).

  1. On page 7, last paragraph, I did not understand the lbif-->lbif example as it seems that nothing is repaired?

Response:  Sorry; there was an error.  Lbif is expected to be repaired as Lebif.  The error has been amended in the text.

  1. After reading the paper, I am left with a conclusion that while some individuals with dyslexia (and sometimes by definition) exhibit poor performance in some phonological tasks, we have no theory of how that would explain dyslexia that we could test. I think this is a very important message to put out.

Response: I sincerely thank the reviewer for this evaluation and comment.

Reviewer 3 Report

The paper is very clear and presents arguments that are timely and needed to move the field forward. I have only a few suggestions that I would like Professor Zoccolotti to consider:

  1. The term dyslexia is used assuming that it means the same thing for different readers. As (1) some definitions include mention of phonological deficits and others do not, and (2) some studies use participant selection criteria that limits participants to those with a phonological deficit, I think it would also be useful for the internal consistency of this paper to provide a definition dyslexia that does not include phonological deficit. I understand definitions come with a whole new set of issues, but I believe anchoring the arguments against the authors definition would still be valuable.
  2. In few places I read the argument that it is well known that individuals/children with dyslexia are impaired in phonological awareness tasks (pp. 2 & 4 at least). This is true only if the sample was selected on the basis of phonological deficit. When samples are selected on the basis on poor reading accuracy or efficiency, the dyslexic groups may still be poorer than the control group but when individual level information is available, up to half of them do not show phonological deficit. These statements should be qualified by "as a group" or "many, but not all" or something of that kind - if not, it perpetuates the misperception of universality of poor performance in these tasks. I think this is also true with categorical perception tasks - when distributions of scores are reported, there is a large overlap between groups with many individuals with dyslexia performing at normal levels. 
  3. In terms of RAN deficits, it may also be worth mentioning that several studies have found a group that has RAN but no PA deficits --> if both are explained by phonological deficit, we need two independent deficits or a well-developed severity argument (and I am not aware of one).
  4. On page 7, last paragraph, I did not understand the lbif-->lbif example as it seems that nothing is repaired?
  5. After reading the paper, I am left with a conclusion that while some individuals with dyslexia (and sometimes by definition) exhibit poor performance in some phonological tasks, we have no theory of how that would explain dyslexia that we could test. I think this is a very important message to put out. 

Author Response

Review of Zoccolotti: Brain Sciences

I find it strange being asked to review a paper that, alongside some positive constructive comments, is largely critical of my Misconceptions paper. For this reason, I don’t think it appropriate for me to make an accept/reject/revise decision. I will leave that to the editors and confine my comments to the substance of this piece.

Response: I like to express my appreciation to prof. Share for his fair position as well as for taking the time to extensively comment on my paper.

  1. 1. The comments about “metaphorical” expressions (ensemble, tip of the iceberg, torrent) I think are unnecessary. Questioning the use of these terms is not addressing the substance of the arguments, just the style of writing. I chose to use colorful terms such as “torrent”, the author of this piece himself (Z.) uses the term “very large”, Berent uses the word “large body”, but no- one including Z. is challenging the substance of my claim that there were a large number of papers (including reviews and meta-analyses) in the period referred to. The substance of the opening argument (and much of the commentary) concerns the “lack of specific predictions’ of the PDH which I address in my second point (below) and argue represents a fundamental misunderstanding. So I recommend that the reference to metaphorical expressions be dropped.

Response: I would like to note that there is a large literature about the use of metaphors in scientific texts (to which I also contributed minimally).  In general, their use is much more spread than one may anticipate.  Metaphors help to effectively convey the meaning and as such have an important communicative value.  In this particular case, the point I wished to make in the original version of the text is that the metaphoric expressions were used in the absence of an explicit formulation of the theory. Thus, I agree with prof. Share that what is most important is the substance of argument and not the form but, in this particular case, the point made in the original version of the text is that the use of metaphors was instrumental in describing a theory in the absence of an explicit formulation of its predictions. I would like to add that my intention was quite specific along these lines and not to make any negative remark on the form.  At any rate, I have substantially revised the text in order to make this point more clearly.  I hope that the new formulation proves clearer.

  1. The author’s main claim is that there is no explicit and testable formulation of the PDH theory. This is incorrect. The PDH has been repeatedly tested and upheld across languages and orthographies around the world and is the reason why the PDH today has achieved such a remarkably wide consensus. In my opinion, there is a fundamental misunderstanding here that rests on confusing two different lines (or aims) of PDH research. The first generation of this work (mostly undertaken over the last 3 decades of the 20th century) aimed to test phonological versus non- phonological accounts of dyslexia. The goal (in most cases) was to challenge the old status- quo which saw dyslexia as a visual-perceptual and/or perceptual-motor deficit (e.g., Satz). We now have literally hundreds of studies showing that the problem (for most but not all dyslexics) is primarily phonological, not visual and not semantic. This was and still is an explicit and falsifiable hypothesis and has been repeatedly supported. So much so, in fact, that it’s all too easy to overlook this first generation of PDH research which today is largely taken for granted. To re-iterate, this first generation was looking at phonology from the “inside-out” inquiring whether the locus of the difficulties is phonological or non-phonological. The author’s (in my view misplaced) criticism about a lack of explicit and testable predictions centers on a second generation of research with a different goal: Those that the author is referring to (e.g., Ramus, Berent, Szenkovits and others) all agree that there is a phonological deficit, their aim is refine the PDH, asking what exactly is the source/nature of the phonological deficit. These researchers are looking within the domain of what is broadly termed “phonological processing.” (There are good reasons why first generation researchers labeled the problem as phonological processing, and not strictly phonological in Berent’s narrower linguistic sense). In their own words....

Ramus et al. (2008) quite emphatically stated that “The first important remark to make is that our results do not challenge in any way the very existence of a phonological deficit (my emphasis). Indeed, our own data attest that our participants with dyslexia have a phonological deficit, as measured in the traditional sense, using for instance spoonerisms, nonword repetition, and rapid naming tasks. So it is not time to abandon the phonological deficit hypothesis, merely to rethink its precise formulation.”

Berent offers a strictly linguistic definition of phonology (phonological grammar) one that is much narrower than the conventional usage of the term “phonological processing”. This term was coined by psychologists to refer to a broad range of tasks – not just speech perception, not just picture naming, not just pseudoword repetition, not just verbal short-term memory but to a set or family of measures that typically create difficulties for dyslexics). Berent reports data showing that source of the speech processing deficits of dyslexia which she accepts, is (primarily) phonetic perception. She too, is not rejecting the PDH but, like Ramus et al. seeks to refine it.

Berent et al., 2013 writes “A large body of research demonstrates that dyslexia is associated with a host of deficits to the processing of spoken language, including abnormalities in the identification and categorization of speech sounds (Blomert, Mitterer, & Paffen, 2004; Brandt & Rosen, 1980; Chiappe, Chiappe, & Siegel, 2001; Godfrey, Syrdal-Lasky, Millay, & Knox, 1981; Mody, Studdert-Kennedy, & Brady, 1997; Paul, Bott, Heim, Wienbruch, & Elbert, 2006; Rosen & Manganari, 2001; Serniclaes, Sprenger-Charolles, Carre ́, & Demonet, 2001; Serniclaes, Van Heghe, Mousty, Carre ́, & Sprenger-Charolles, 2004; Werker & Tees, 1987; Ziegler, Pech-Georgel, George, & Lorenzi, 2009), and deficits in talker identification (Perrachione, Del Tufo, & Gabrieli, 2011) and in discriminating speech from nonspeech (Berent, VakninNusbaum, Balaban, & Galaburda, 2012). Moreover, many of these deficits are already evident in early development, well before reading is acquired (van Herten et al., 2008; Leppa ̈nen et al., 2002).”

Szenkovits et al. (2016) are also unequivocal in their support for the PDH“....”focusing on the majority of people with dyslexia who do have a phonological deficit, the present study aims to further investigate the nature of that deficit”, p.317”. ...these dyslexic adults still exhibit the typical phonological deficit as measured by phonological awareness, verbal short-term memory and rapid automatic naming tasks. Thus, it is suggested that the explanation for their phonological deficit must lie elsewhere than in their phonological representations and grammar.” Even the titles of the two papers - Exploring Dyslexics’ Phonological Deficit....I and II makes clear that they are not repudiating the PDH.

All these researchers are adherents of the PDH, second-generation researchers seeking to further refine it, by pinpointing the precise nature of the deficit at a finer level of resolution. Within this line of work which, for want of a better term, I am calling second-generation within-phonology research, Z. is correct in saying that we (at present) lack specific and explicit predictions of the type that Ramus, Szenkovits and Berent are seeking. For example, Ramus and Ahissar address the hypothesis of degraded phonological representations (one version of the PDH), asking if it’s “noisy, sparsely coded, underspecified, with poor spectral or temporal resolution”, or if not the representations themselves, perhaps their accessibility? Berent was looking to determine what is the source of the speech processing deficits - phonetic perception? phonological grammar, etc?

And there are many other dyslexia researchers who, while accepting the PDH, are aiming to provide a higher-resolution instantiation or specification of the hypothesis. In this sense, the author is missing the forest in his focus on specific trees.

Response: There is certainly a clear difference in the way prof. Share and I look at this overall question.

One key argument of prof. Share is that a first generation of studies (in the late nineteen century) set the phonological versus non-phonological account of dyslexia.  Then, a subsequent generation of studies (in the last two decades) set out to “refine the PDH, asking what exactly is the source/nature of the phonological deficit”.  In this perspective, the phonological interpretation of dyslexia is granted by the first set of studies and only some specifications of this interpretation are left for current research.  I have some personal difficulties in following this argument although I sincerely do not wish to have a misunderstanding of this. It would seem to me that one cannot state that an interpretation is correct in spite of the results of further research.  It is conceivable that phonological processing has to do with dyslexia (and this would be the outcome of the first-generation studies).  However, proposing that this can be expressed in terms of a “theory” indicates that one is ready to specify the conditions (in terms of type of stimuli, tasks, individual differences etc.) under which one expects children with dyslexia to fail and those in which no failure is expected. I realize this is talking about “trees”, but this is what theories are about, i.e., the explicit formulation of testable predictions.  If this is not the case, one is left with the impossibility to know whether the available evidence fits or not with the theory (put in other terms the theory is not falsifiable).  Is the observation that phonological grammar is spared in children with dyslexia inconsistent with the theory or not? Or the finding that the performance of children with dyslexia naming pictures is the same as their reading proficient peers? I restrained myself from drawing general conclusions from these null results.  Rather, I limited myself in noting that it is extremely difficult to frame these and several other results stemming from current research as the theory is not explicit in its predictions.  Thus, one is left with the need to interpret each single negative finding in a post-hoc way.

I would like to stress that in this argument I am trying to move, using a metaphor, in a “narrow road”: the aim of my paper is not that of making an extensive evaluation of the PDH theory (which would have requested a considerably larger analysis and a much more extensive rendition).  More selectively, I raised the specific point that the PDH theory is largely underspecified which impedes evaluating the way experimental results fit with this theoretical proposal.  In his point 5 (see below), prof. Share acknowledges that he did not spell out an explicit formulation of the PDH as he took it for granted.  I am certainly not questioning this presentation choice. It is also true that a large number of papers make reference to the PDH theory of dyslexia; so, this certainly does not come as unknown.  However, the argument I made in the paper is that it might prove extremely complex to formulate the theory beyond a general statement in a way that it can account for the large heterogeneity of currently available findings.

Prof. Share correctly states that there is a large consensus over the PDH theory.  Thus, several authors (such as Ramus, Szenkovits and Berent) in different forms express their agreement with this theoretical perspective (in spite of having themselves produced results at least in part incongruent with the expectations). I certainly do not disagree with this point and in various parts in the paper I have stated that the PDH theory has a wide consensus.  Indeed, this is one of the reasons for the interest in focusing on its examination. However, (as stated in the title of the paper) it is not the consensus or success among researchers which is critical in the evaluation of a scientific theory but the capacity to provide an explicit framework of predictions on the target behavior.

  1. 3
    . Alongside the positive findings, the author discusses some “negative” findings that he believes show that the PDH is vague and poorly formulated to the point that it is unfalsifiable. These include categorical speech perception, picture the PDH lies not in any one single measure (not even PA), but the consistent and well-replicated fact that naming and RAN. At the level of individual measures such as categorical speech perception, picture naming and RAN, many of the same concerns have been voiced in the past by various authors including myself (see, Share, 1995, p. 181 with regard to speech perception, and p. 183 on discrete versus serial picture naming (RAN)). Focusing on specific trees (measures), the author, once again, is missing the big picture which is that no single measure is definitive –it’s the collective strength of these measures that is crucial. By focusing on single tasks the author is missing the point that the strength of on the set of phonological processing measures, most (but not all) dyslexics perform poorly. (And, it’s important to note that there are many other phonological processing measures besides these three, including phonological short-term memory, articulatory awareness, PAL – paired-associate learning, gating tasks and more).

The author rightly cites the (legitimate) concerns of many dyslexia researchers about imprecision of behavioral measures and the difficulties (indeed impossibility) of developing “clean” behavioral measure and the problems of making inferences about unobservable underlying constructs. Perhaps it’s here that we should be looking to the neurobiologists for clarity in the future. (Clearly it is the hope of the editors of this special issue to get some traction here on the neuropsychological basis of dyslexia, but it’s still early days in brain research because the brain is more complex than we thought, more plastic than we thought and still too poorly understood, so the promise of neurobiology still lies well in the future as witnessed by the fact that the number of neuro-theories seems to match the number of research groups working at this level of explanation.)

Personally, I doubt that our behavioral measures will be able to get us much beyond the basic first-generation level of resolution. Maybe we are asking too much of our imprecise behavioral instruments? Method variance is (unfortunately) often a major factor (and typically under- appreciated) in patterns of correlation and between-group differences. Hence the necessity for multiple measures and the importance of recognizing the dimensional nature of behavioral disorders such as dyslexia (i.e., degrees of severity, Stanovich, 1999; Snowling, 2020). To be classified as ADHD, a child must show a minimum number “symptoms” - 6/18 according to DSM. No single “symptom” on the Connors rating scale is definitive it’s the collective picture. The same I believe applies to dyslexia. The fact that dyslexia, like ADHD is not categorical but dimensional means that some dyslexics exhibit some of these “symptoms”, others more, without any dyslexic’s individual profile of “symptoms” being necessarily identical. Again, it’s the big picture of depressed performance across many, most, or even all of these phonological processing measures that counts.

Response: In the first part of point 3, prof. Share underscores the importance of considering not only individual measures (such as categorical speech perception, picture naming and RAN) but the “collective strength of these measures”.  Otherwise, “by focusing on single tasks the author is missing the point that the strength of the PDH lies not in any one single measure (not even PA), but the consistent and well-replicated fact that on the set of phonological processing measures, most (but not all) dyslexics perform poorly”.

In our own research project in the last years, we have focused on examining global components in children with dyslexia (referring to models of individual processing such as the difference engine model or DEM by Myerson et al. 2003).  So, I am personally sympathetic to the idea of looking at global component in a large-scale perspective.  However, here the point seems different.  If, for the sake of discussion, we take the measures quoted above (i.e., categorical speech perception, picture naming and RAN) there seems to be no clear way to put together these pieces of information.  Clearly, these tasks refer to different underlying processes. As briefly summarized in my note, some processes indicate very limited or no impairment in children with dyslexia (such as picture naming) while others indicate more consistent impairments (e.g. PA tasks).  How should they be considered unitarily?  In other terms, we have a cluster of different tasks/conditions referring to a cluster of different processes.  It does not seem straightforward to me how this information can be considered unitarily.  Prof. Share adds that there are further conditions which may bring relevant information (namely phonological STM, articulatory awareness, PAL, gating tasks.  Similar considerations apply to these conditions.  It does not seem simple to find ways to consider data from these tasks unitarily and, to the best of my knowledge, this has not yet been done.

The third par of reviewer’s point 3 develops this question with regard to the stability or reliability of measures: “Personally, I doubt that our behavioral measures will be able to get us much beyond the basic first-generation level of resolution.” If I understand it correctly, the case of ADHD exemplifies the idea that, in a dimensional perspective, one could envisage that children with dyslexia fail in a proportion of cases/conditions although it is not possible to anticipate specifically which ones are those in which this will occur (because of the low resolution of these measures).

I have two comments on this position.  First, the cases of ADHD and dyslexia may not be so similar.  In the case of ADHD (on which I admit limited knowledge), the symptoms in a given scale (either inattention or hyperactivity) refer to relatively homogeneous behaviors (though clearly there is a variation in severity and the threshold seems to refer to this).  In the case of dyslexia, at least in the example used above, measures refer to quite different processes.  Thus, it is not only a matter of overall severity but also one of deciding among different measures (i.e., the trees in the metaphor) and underlying processes (i.e., their roots).  Second, if one wants to take this stand, it would seem important that this is explicitly stated in the formulation of the PDH theory.  Put in other terms, one could certainly express the idea that children fail in a variety of tasks loosely falling into the phonological category even though it is not possible at present to anticipate which ones (but only predict that children with dyslexia will fall in a sizeable proportion of tasks).  To my knowledge, this point has never been made so explicitly.  I would certainly be very happy if my paper helps in stimulating a more explicit formulation of the PDH theory.  I made this latter point more explicitly in the last sentence of the revised text.

  1. 4. I think the section on proximal/distal has some valuable discussion, and also the final section – but again, it is essential to distinguish between research aimed at refining the PDH as opposed to (first-generation inside-out) research aimed at comparing phonological to non-phonological accounts.

Response: I thank prof. Share for his positive comment on this part of the manuscript. 

As stated above, I have some difficulties in fully understanding why first- and second-generation studies should be treated differently and separately.

  1. 5. It is true that my Misconceptions piece did not spell out an explicit formulation of the PDH (I took this for granted, although I did add the new writing systems angle to show why phonology is important to learning to read), perhaps this could be added in some future reply/commentary if the editors are interested.

Yet another issue is heterogeneity (for future commentary) – individual differences and subtypes. This piece (and mine too to some extent) tends to treat dyslexics as a single undifferentiated group. If there are subtypes with different profiles then group means will tend to show marginal or “subtle” deficits.

Response: I personally feel that it would be quite interesting if prof. Share would provide a further paper on the PDH theory.  As stated by him, this of course rests upon the evaluation of the Editor of the Special Issue.

I agree that individual differences (and subtypes) are an important issue and I have made a limited reference to this point in the revised version of the manuscript.  Still, to a large extent this important problem goes beyond the scope of the present paper.

Round 2

Reviewer 1 Report

After reading the author’s response to my comments 1 and 2, I realized that most concerns I had were due to the expression ‘lack of explicit predictions’, which I interpreted as non-testable/hard-to-test predictions (like, e.g., ‘the superego exists’). But now I believe the author meant something different, namely ‘explicit theory’ or even ‘proper theory’ (a theory referring to components and mechanisms) and ‘precise predictions’, these referring to predictions based on all evidence accumulated so far. So, I think we had a language issue.

The response to my comment 3 was satisfactory.

Because it is possible that I am being too strict (or even wrong) regarding the terminology and its implications to the overall structure of the paper, I will not insist in this terminological debate. The paper has interesting ideas, it feeds an important ongoing debate on dyslexia and is generally well-written. So, I will endorse it, hoping that other readers will not be intrigued with the idea of ‘lack of explicit predictions’ as much as I was.